# OpenReview forum: "Interpreting Language Reward Models via Contrastive Explanations"
_ICLR.cc/2025/Conference — ICLR 2025 Poster_

### Official Review · Reviewer_T2w6 · 2024-11-01

**Soundness:** 3
**Presentation:** 3
**Contribution:** 4
**Rating:** 8
**Confidence:** 3

**Summary:**

This paper studies the interpretability of reward models through contrastive explanations. It proposes a novel pipeline of getting high-level attributes, computing meaningful textual perturbations, and categorizing into counterfactual (CF) and semifactual (SF) explanations. Then it performs case studies on how to use the CFs and SFs to interpret reward model decisions.

**Strengths:**

Overall i like this paper!
* The problem of RM interpretability is important.
* The whole pipeline of generating CFs and SFs make sense
* The case studies demonstrate the usefulness of the generated explanations very well

**Weaknesses:**

* I am a little skeptical of the high-level attributes concluded by the model. The attributes are generated per sample, and then aggregated to find the top attributes. Then does the LLM give consistent attributes (e.g. do they consistently use the exact word of avoid-to-answer?)? Even if an attribute shows up in LLM's summarization for many samples, does it always mean the same thing (e.g. attribute complexity might mean good/bad things)?
* Does the prompting in step 2 always work as intended? In Fig 4, it seems some responses are modified almost completely --- does it mean the LLM doesn't faithfully follow step 2 prompt of only modifying the identified word?
* Step 2 prompting requires the modification to be both better/worse, and also better/worse in terms of targeted attribute. Why do we still need the first condition?
* Are there more compelling evidence for the usefulness of the paper? E.g. some reward models with known issues (e.g. length bias), and the generated data can identify that?

**Questions:**

I put my questions in the weaknesses

---

> ### Author Response · Authors · 2024-11-21
> **Responses to reviewer T2w6**
>
> We thank the reviewer for the constructive feedback. Please see our common response to all reviewers above, for details of our updated revisions and responses to two specific questions. We provide detailed responses to the questions below:
>
> ### **Response to weakness 1:**
>
> Please note that the process for determining the attributes can be very flexible, and in our case, it is semi-automated. Indeed the attributes generated by the LLM are not always consistent. In our implementation, we first obtain a dictionary of the form {attribute name: number of times occurred}, then we manually group similar ones, e.g. harmless and harmful, too-long, too-short, and verbosity. We further clarify the meaning of each attribute by providing a description to the LLM (see Appendix A), which should mitigate the ambiguity issue you mentioned.
>
> ### **Response to weakness 2:**
>
> Our Step 2 prompt says “changes made to response A should be *centered around* the following words: {relevant words for this attribute identified in Step 1}”, but does not ask LLMs to *only* change those identified words. The LLM might change the whole response for two reasons. First, in Step 1, the LLM sometimes marks nearly every word in the sentence as relevant to the attribute, therefore in Step 2 the LLM could change the whole sentence. Second, in the implementation, we have an error-handling mechanism to ensure experiments execute at scale: if errors occur in Step 1, then skip Step 1 and directly execute Step 2 while allowing the LLM to change the whole sentence. The second case very rarely occurs. According to our observations presented in Appendix D, our current configuration in Step 2 (asking LLM to center the changes around relevant words identified in Step 1) works well because perturbations produced by such prompt are: (1) more semantically and syntactically closer to the original responses than the ones produced by the “pass” prompt (not limiting the extents of changes), and (2) more semantically and syntactically distant to the original responses than the ones produced by the “only” prompt  *=(asking LLMs to only change those identified words).
>
> ### **Response to weakness 3:**
>
> Perturbing the original response to become better or worse will depend on whether this response is the chosen or rejected response. We perturb the chosen response to become worse and perturb the rejected response to become better, to hopefully flip the RM preference.
>
> ### **Response to weakness 4:**
>
> We have additionally included more qualitative results revealing a new failure case of larger 8B RMs. Please see the common author response and Appendix G in the paper for more details.

---

### Official Review · Reviewer_CGue · 2024-11-02

**Soundness:** 3
**Presentation:** 4
**Contribution:** 3
**Rating:** 8
**Confidence:** 4

**Summary:**

The paper presents an analysis technique for reward models. By perturbing input examples with LLMs along pre-determined attributes to generate counter-factual (CFs) examples (that flip the preference judgment of a reward model by modifying an attribute) and semi-factual (SFs) examples (examples that retain the label despite editing an attribute), the authors are able to obtain quantitative (robustness to perturbation) and qualitative (which attributes are important to the preference judgment) signal about reward model judgments.

The main contribution is the analysis technique itself, with experiments comparing the quality of CF/SFs generated by the proposed prompting technique against baseline methods of perturbation to demonstrate the quality of explanations generated, and a case study comparing three open-source models to use the method to identify differences in their performance qualitatively.

**Strengths:**

1. The proposed technique is clean and simple, draws from existing literature on SF/CFs in XAI, and importantly is general enough to work for any RM (L.132-134), making it a useful tool for analysis. This is particularly the case when you might not have access to the training data of some black-box RMs and can yet obtain qualitative insights on their performance.

2. The main contribution is the analysis technique. Section 3 makes a good case that the proposed prompting technique is an effective way to generate good CF/SFs and Section 4 then shows how this can be used to interpret RM behavior. The flow is easy to follow and the experiments validate the authors' claims.

3. L.405-411 - The experimental setup chosen for Section 4.1 is clever because we have ground truth knowledge that the Deberta-v2 reward model was the only one trained on the _harmless_ dataset, so it is more sensitive to perturbations that change the 'harmless' attribute. This validates that the behavior identified by the proposed analysis technique is 'correct' in this case. A very interesting follow-up would be introduction of perturbation attributes that are not directly related to the training task of the RM, and then this technique can identify biases in model performance.

**Weaknesses:**

1. While the experiments generally back up the claims made, I think there is scope to more rigorously test the robustness of the proposed technique. See Q2, 3, and 7 below for specific concerns. Another concern is that the generation method for attributes looks at examples from the test set. This is fine, but it grounds the perturbations to 'expected' ways and doesn't elicit any 'surprising' biases from RMs. This is easily remedied of course, simply by sampling attributes along different ways, but it would be interesting to see if the generated SF/CFs are sensitive to the attribute distribution.

**Questions:**

1. L.42 - Can you clarify the distinction to (Zeng et al. 2024) and (Park et al. 2024)? My understanding is that these papers make directed perturbations based on a specific attribute, while the proposed technique looks to analyze model performance via a set of targeted perturbations (SF/CFs). I don't see this as a weakness per se, it's more that L.59 mentions that there have been no XAI techniques explored for reward models but the way L.42 has been written contradicts this.

2. One unanswered point is about the sensitivity of the analysis technique to the choice of LLM used to generate CF/SFs. Here the authors use GPT-4o. I think it is important to test because the main contribution is the analysis technique itself, so we should ablate the sensitivity to other (less powerful/open-weight) LLMs since not everyone might have access to GPT-4o.

3. L.186 - Can you comment on the sensitivity of the method to the selected high-level evaluation attributes? Are there patterns of evaluation attributes for which the generated CF/SFs are less effective at flipping the label - in these cases, how do we decide if this is because the RM is robust to perturbation along this attribute or if the LLM generating the counter-factual is not able to correctly edit the response based on that attribute.

4. L.262-265 - In addition to the fraction of test comparisons where at least one valid CF/SF is found, why not differentiate between cases where multiple valid CFs exist as examples where there is more brittleness in the RM?

5. L.706-706 - This method to elicit attributes from the test set is similar to techniques that use LLMs for qualitative coding so it might be helpful to ground the method in similar works [1,2,3].
[1] Xiao, Ziang, et al. "Supporting qualitative analysis with large language models: Combining codebook with GPT-3 for deductive coding." Companion proceedings of the 28th international conference on intelligent user interfaces. 2023.
[2] Dai, Shih-Chieh, Aiping Xiong, and Lun-Wei Ku. "LLM-in-the-loop: Leveraging Large Language Model for Thematic Analysis." Findings of the Association for Computational Linguistics: EMNLP 2023. 2023.
[3] Chew, Robert, et al. "LLM-assisted content analysis: Using large language models to support deductive coding." arXiv preprint arXiv:2306.14924 (2023).

6. In Table 1, is there a reason for the drop in performance between the coverage on CF and SFs that were generated perturbing 'Both' responses than just "Chosen" or "Rejected"? Uniformly these scores are lower.

7. Regarding the qualitative analysis in Section 4, it might be interesting to introduce some 'control' attributes that are not meant to flip the label of the plot to provide some context to the 'treatment' attributes selected. For instance, I imagine Fig. 3 would be improved with such additional attributes, and I imagine if an RM flips a label on these attributes, then it's a method to identify biases in the judgments.

8. L.324-326 - While it is understandable that the generated CF/SFs are more similar to the original response which likely costs diversity, I think you have a better justification for the method, than the using of the 15 different attributes is just pure efficacy i.e. the prompting method generates valid CF/SFs at a higher rate than the baselines, this itself is sufficient.

---

> ### Author Response · Authors · 2024-11-21
> **Responses to reviewer CGue**
>
> We thank the reviewer for the constructive feedback. Please see our common response to all reviewers above, for details of our updated revisions and responses to two specific questions. We provide detailed responses to the questions below:
>
> ### **Response to weakness 1:**
>
> Please see the common response about the list of attributes.
>
> ### **Response to question 1:**
>
> The main difference is the target of the research. These existing works aim at evaluating the RM’s ability to capture various types of biases by curating datasets and testing RMs’ performance on them, while our approach aims at explaining any binary comparisons made by an RM (although by using our explanation approach we could perform similar analysis to theirs). Another notable difference is in the methodology. They perturb one response along certain directions and compare between those perturbations, while we perturb two responses in a binary comparison and compare the perturbations of one response to the other original response. Also, please note that our explanation method (Figure 1) could be instantiated with any customised list of attributes. In that sense, our proposed method complements these existing works.
>
> ### **Response to question 2:**
>
> Automated text perturbation procedures are indeed sensitive to the LLM used. For example, [Bhattacharjeea et al., 2023] and [Sen et al., 2023] compare various LLMs for generating perturbations for counterfactual explanations (in classification tasks) and augmenting training data, and it is evident that their quality (label flip rate in classification, semantic and syntactic distances to original responses, diversity, etc.) are affected by using different LLMs. Overall, the finding is that less capable LLMs will generate perturbations that are more similar to the original text but less frequently change the model prediction result. Also, better LLMs will more likely generate perturbations that match the meaning of the specified attribute and are more natural texts. Therefore, we recommend a user of our method to use the most capable LLM available (in our case GPT-4o) to maximise the usefulness of our method. Additionally, the quality of any perturbation-based XAI methods will depend on the perturbation quality. In our case, the formalisations of the explanation method are agnostic to the generator of the perturbations.
>
> ### **Response to question 3:**
>
> Please see the common response about the list of attributes.
>
> ### **Response to question 4:**
>
> The quantitative evaluations in our work (Section 3) follow the standard practice in the contrastive explanations literature, and the coverage metrics aim to test an explanation generation method's ability to obtain valid explanations, instead of evaluating specific aspects of the RMs. Discovering the sensitivity of RMs to the list of attributes is a target of the following qualitative analysis in Section 4.
>
> ### **Response to question 5:**
>
> Thanks for this comment, we added references to them in the revised paper in Appendix A.
>
> ### **Response to question 6:**
>
> Yes there’s a reason for that observation. It could happen that for one comparison, CFs (SFs) are found among the perturbed chosen responses but are not found among the perturbed rejected responses, which contributes to the lower coverages when checking whether CFs are present for both original responses.
>
> ### **Response to question 7:**
>
> We have additionally included more qualitative results revealing a new failure case of larger 8B RMs, please see the common author response and Appendix G in the paper for more details.
>
> ### **Response to question 8:**
>
> Thanks for this comment!

---

> > ### Comment · Reviewer_CGue · 2024-11-23
> > **Rebuttal acknowledgement**
> >
> > Thanks for the response, both for adding experiments and for answering the questions! I think the added experiments in Appendix G is a valuable addition to the paper.
> > Before finalizing my score, I have two more clarification questions:
> >
> > On the response to Question 2: I buy the argument made in the response, and also that all perturbation-based analysis would suffer from the same limitation, but I think I would still like this explicitly mentioned in the paper.
> >
> > I'm not sure I follow the response to Question 6. Does this mean that 'Both' refers to examples with perturbations in 'Either' response? Perhaps you could elaborate.

---

> ### Author Response · Authors · 2024-11-24
> **Author response**
>
> Thanks for acknowledging our responses and, again, for the very detailed comments.
>
> We agree that the sensitivity to the LLMs that generate the perturbations should be mentioned in the paper. We further added comments about this in lines 284-286 in the updated revision.
>
> About Question 6, sorry if that was not clear. Here is a more detailed explanation: the "Both CF (SF) Cov." evaluation metric checks whether or not for each test comparison (x, y+, y-) at least one CF (SF) is found for both y+ and y-.
>
> Assume we have a dataset with three test comparisons, for comparison 1 a CF is found for y+ but not for y-; for comparison 2 a CF is found for y- but not for y+; for comparison 3 a CF is found for both y+ and y-. Then, for this dataset, the Chosen CF Cov. is 0.67, the Rejected CF Cov. is 0.67, but the Both CF Cov. is 0.33.
>
> The same reasoning applies also to evaluating SF.

---

> > ### Comment · Reviewer_CGue · 2024-11-25
> > **Acknowledgement of Response**
> >
> > Got it, this makes more sense now. I've updated my review, and thank the authors for engaging with my concerns in a comprehensive manner despite the short turn-around time.

---

### Official Review · Reviewer_WCX9 · 2024-11-03

**Soundness:** 3
**Presentation:** 3
**Contribution:** 3
**Rating:** 6
**Confidence:** 3

**Summary:**

The authors propose a two-step approach to generating diverse perturbations of language completions, and show that this method can be used to improve the diversity of contrastive explanations of reward model outputs.  (i.e., perturbations that result in pairs of counterfactual or semifactual paired completions that can then be used to explain a reward model's behavior).

They validate their approach both quantitatively and qualitatively and show how it can give insights into reward model behavior, including the sensitivity of the reward model to perturbations along a set of specified axes, both at a local and global level.

**Strengths:**

I am not very familiar with the XAI literature or its application to reward models, but I have not seen this particular approach before and believe the methodology is novel (but see weaknesses for some requests on related work). I think researchers who work with reward models will generally be interested in ways of explaining the reward model predictions, and so I believe the topic and proposal in this paper meets the significance threshold for a major conference (but see weaknesses re: chosen RMs).

Overall the paper was well written, and the figures are generally high quality, so I believe it has sufficient writing polish for publication. See weaknesses for some nits.

While the method itself is relatively straightforward (the authors cite a couple works that have done similar things in other contexts), it is demonstrated to be effective. The experiment design is appropriate, and the authors are testing the right things. Despite the limited RMs tested and the use of binary preference reversals (pref flip rate) instead of cardinal rewards (see weaknesses), I believe the method itself is reasonable, and the experiments provided sufficient support for the approach.

**Weaknesses:**

Related work: At lines 179-181, a few related works are mentioned that could alternatively do perturbations or CFs for reward model explanations. These are not expanded upon in the text, and they do not seem related to the baselines at lines 277 - 284. Is this because they do not apply at all, or how do the baselines relate to past work? This casts some doubt on how the present work fits into the literature.

Conceptual: It seems the finite attribute list limits the kinds of explanations that the proposed method can provide. For a given part of completions it seems possible that an attribute that doesn't quite relate to some of the listed attributes (or related to multiple listed attributes) would be the main causal factor. For example, one completion might be preferred because it uses a more clever metaphor than the other completion. I'm not sure how this would fit in any category, and it seems like perhaps a dynamic approach or wildcard attribute might be better in some respects (I realize this might conflict w/ the experiments that rely on ranking attributes).

Conceptual: It's unclear to me why we would prefer Preference Flip Rate (PFR), which is an ordinal measure, to the cardinal measure provided by the rewards themselves. For comparability when rewards are of different magnitude, you can always normalize by either the mean reward, or the gap between chosen and rejected. A cardinal approach seems strictly superior to me, and no real justification is provided for considering CF/SF to cardinal shifts in rewards (I would wager than the authors use flips because of the prior work that generates CF/SF contrastive explanations).

Significance: The tested RMs are all extremely small / poor by today's standards, and would not be used in any real context. 7-8B parameter RMs are many times stronger and more relevant, and are easily runnable on a single 24GB GPU, so I would not expect there to be a compute barrier here. So it's not clear to me that the positive results obtained for these RMs will carry through to modern ones. This limits the present significance of the results, and I would strongly encourage the authors to run experiments with one or more stronger RMs.

Nits:
- L419/420 mejntions that you compare global/local rankings--- are there supposed to be results for this, or is this just to produce the Figure 4 example
- Figure 4 caption could be significantly improved. It took me a second to figure out how to read it / what it was presenting (mind you, I was reading in black and white so couldn't see the color difference between the dashed lines, but it would still be good to explain them in the caption).

**Questions:**

Please see weaknesses. The significance one would provide the largest improvement, but perhaps hardest to address. The related work one should be easy to address and therefore high ROI.

---

> ### Author Response · Authors · 2024-11-21
> **Responses to reviewer WCX9**
>
> We thank the reviewer for the constructive feedback. Please see our common response to all reviewers above, for details of our updated revisions and responses to two specific questions. We provide detailed responses to the questions below:
>
> ### **Response to weakness on related work:**
>
> The existing works mentioned in lines 179-181 target classification tasks in textual data. They either use a custom fine-tuned LLM (e.g. [Wu et al. 2021]), or directly prompt LLMs like GPT-4, to generate counterfactual texts for one piece of input text. Our key novelty is proposing the framework for explaining pairwise comparisons, which involves two pieces of text and is therefore substantially different to existing works. Although the part about generating perturbed texts is similar, no existing works directly apply to explain RM preferences. In fact, no existing works apply XAI methods for RMs.  In our baselines, we adapt [Wu et al. 2021] (PJ) and naïve prompting for LLM (RP) into our explanation framework.
>
> ### **Response to weakness on conceptual 1:**
>
> Please see the common response about the list of attributes.
>
> ### **Response to weakness on conceptual 2:**
>
> We consider the flipping preferences as central to our definition of contrastive explanation, which aims to explain why an RM prefers one response over another. Analysing cardinal reward changes does not directly serve this explanatory purpose, but analysing flip rates does. Also, an RM's predicted reward for a response, if outside the scope of a comparison, could be difficult to normalise and interpret due to the pairwise training procedure. The notion of binary preference flipping is more straightforward to understand, especially for non-experts. Nonetheless, the presented analysis pipeline is independent from our core explanation generation method, so alternative analyses using the cardinal measures could be conducted in future work.
>
> ### **Response to weakness on significance:**
>
> We have additionally included more qualitative results revealing a new failure case of larger 8B RMs. Please see the common author response and Appendix G in the paper for more details.
>
> ### **Response to Nits 1:**
>
> This is part of obtaining the representative local example that matches the global pattern. In the scope of this paper, they are to produce Figures 4 and 5.
>
> ### **Response to Nits 2:**
>
> Indeed, we have added more descriptions in the revision to make the figures clearer.

---

> ### Author Response · Authors · 2024-12-02
> **Follow up author response**
>
> As the discussion period is closing soon, we're wondering if the reviewer has any final questions or comments. Please note that we have updated the paper with revisions addressing the reviewers' initial comments, discussed in the "Common response to all reviewers". Thanks!

---

### Official Review · Reviewer_sH3s · 2024-11-05

**Soundness:** 3
**Presentation:** 3
**Contribution:** 2
**Rating:** 5
**Confidence:** 3

**Summary:**

The work proposes looking for counterfactual (CF) and semifactual (SF) examples to explain reward models. They prompt language models in a two-stage manner, where an LLM is required first to identify words related to a list of attributes and then to perturb these words to flip the reward prediction. Quantitative experiments examine the CF and SF discovery success rate (CF and SF coverage) and reveal the proposed method mainly improves CF coverage at the cost of compromise in SF coverage in some cases. Perturbed sentences in the proposed method also have a lower syntactic and semantic distance to the original sentence. Some qualitative analyses are also done.

**Strengths:**

1. A novel prompting-based counterfactual and semifactual example generation method for explaining reward models
1. Experiments do show some merits of the proposed method, such as generally improved CF success rate (CF coverage).

**Weaknesses:**

1. The work's main motivation, "More transparent RMs would enable improved trust in the alignment of LLMs," is mostly intuitive and needs more explanations. Even if reward models are totally transparent, how they translate to more transparency in the trained policy model is still not clear, as the policy LLM is still a black box model. Math question training data, as an extreme case, often comes with a more rigid threshold of logical consistency in its data and does not translate to improved reasoning transparency of trained LLMs on that data.

1. How the constructed CF and SF examples are "minimally perturbed" and "maximally perturbed" ones are less reflected in the construction method. Did the authors encourage LLMs to increase/decrease the perturbation ratio on an example until they cannot anymore?

1. The perturbation method relies on a subjectively defined list of categories, which limits the scope of explanation in reward model behaviors. The reward model could vary its prediction due to other factors that humans might not think of, which poses a larger threat to its trustworthiness.

**Questions:**

1. Are there any messages we can take away from the qualitative analysis to improve the trustworthiness of reward/policy LLMs? Existing findings are more or less trivial and superficial, and it would be better to provide more insights that either deepen our understanding of reward models, or improve transparency of policy models.

---

> ### Author Response · Authors · 2024-11-21
> **Responses to reviewer sH3s**
>
> We thank the reviewer for the constructive feedback. Please see our common response to all reviewers above, for details of our updated revisions and responses to two specific questions. We provide detailed responses to the questions below:
>
> ### **Response to weakness 1:**
>
> In this work, our aim is to interpret the preferences made by RMs, which can be used for LLM alignment both at training time (in RLHF) or at inference time (e.g. best-of-N sampling). In the latter case, RMs are used standalone, and better-understood RMs make more trusted comparisons over given LLM responses. In the former case, if the final LLM produces poorly-aligned answers, with existing methods it would be difficult to know whether the LLM has correctly optimised a wrong reward, or it has wrongly optimised a correct reward (e.g. reward hacking - the LLM exploits loopholes in the RM to maximise reward). We see our proposed explanation generation method as one valid starting point towards investigating these problems. Meanwhile, more understanding of RMs has been strongly advocated in the literature (see lines 31-45 in the paper), and our method targets this direction.
>
> ### **Response to weakness 2:**
>
> Mentioning minimally and maximally was to introduce intuitions behind how contrastive explanations are traditionally useful in a tabular data domain, where explicitly computing a minimum or maximum distance (e.g. L1, L2 distances) perturbation of the input is possible. Our proposed way of generating perturbed responses is in line with existing works of generating counterfactuals in textual data for classification tasks, where directly computing high-quality explanations through minimising or maximising distances with gradient descent (as done in tabular and image data) is not applicable because the final perturbation would generally not be well-formed sentences. We agree that iteratively prompting LLMs to increase or decrease the extent of perturbation could be an interesting future extension of our method.
>
> ### **Response to weakness 3:**
>
> Please see the common response about the list of attributes.
>
> ### **Response to Question 1:**
>
> We have additionally included more qualitative results revealing a new failure case of larger 8B RMs; please see the common author response and Appendix G in the paper for more details. Apart from that, we see the qualitative result about model v2's sensitivity to harmful contents as a useful validation, because the fact that v2 was trained to detect harmful contents was not visible to our method. We also demonstrate the usefulness of our method for obtaining more insights into why an RM prefers one response over another in Figure 4 by validating desirable RM behaviours and revealing unexpected behaviours.

---

> > ### Comment · Reviewer_sH3s · 2024-11-25
> > **Reviewer Response**
> >
> > Response to weakness 1 and Question 1: I think this is a good point, and I encourage clarifying this in the introduction.
> >
> > Response to weakness 2: The answer of how examples are "maximally" or "minimally" created is still less clear to me, either in the paper or in the rebuttal. These words are misleadingly used in the paper, making the definitions of "semifactual" and "counterfactual" also less rigorous.
> >
> > Response to weakness 3: The response is not totally satisfactory but seems properly justified. I also agree that this is more of a limitation than a weakness of the paper.

---

> > > ### Author Response · Authors · 2024-11-26
> > > **Author response**
> > >
> > > Thanks for the insightful comments! About weakness 2:
> > >
> > > We only take inspiration from contrastive explanations in tabular classification where "minimally and maximally perturbing" are often explicitly specified in the formulation for finding explanations, and do not claim in the paper that they are properties of our method.
> > >
> > > We agree with the comment that the two words could be misleading. We therefore removed it in the most recent paper revision.

---

> ### Author Response · Authors · 2024-11-25
> **Follow up author response**
>
> As the discussion period is closing soon, we're wondering if the reviewer has any final questions or comments. Please note that we have updated the paper with revisions addressing the reviewers' initial comments, discussed in the "Common response to all reviewers". Thanks!

---

### Author Response · Authors · 2024-11-21
**Common response to all reviewers**

We thank the reviewers for their helpful and insightful comments.

We uploaded a revised version of the paper addressing the reviewers' comments. Specifically, we made the following modifications:
- We added a new Appendix G with more qualitative results on larger 8B RMs, and its associated descriptions in lines 336-338 and 467-469.
- We improved the clarity of Figures 4 and 5 by labelling each perturbation as CF or SF in the figures and adding relevant captions.
- We added three more references in Appendix A linking our way of finding the list of attributes to the literature of thematic analysis of texts.
- We additionally identified a minor typo in notations in lines 147-149 and made corresponding corrections in Section 2.2.

Two common concerns were raised by all reviewers. We therefore provide a detailed response to them below. For other questions, please see our individual responses to each reviewer.

### **1. We added a new Appendix G with more qualitative results on larger 8B RMs.**

Following the reviewers' comments, we additionally conducted one investigation with two state-of-the-art 8-billion-parameter RMs, and identified a new failure case using our explanation method. One new RM is a customised Llama-3-8B model on the OffsetBias dataset from [Part et al., 2024], and the other one is Skyworks-8B RM from [Liu et al., 2024]. Both are among the top-performing models on the RewardBench benchmark.

Starting from the undesirable behaviours of the v2 RM discussed in Section 4.2.1, we tested two 8B RMs on the same local example and discovered that they do not improve on all the failure cases of v2. Further, based on the observation that v2 prefers the response "What the hell do you mean by that?" than "What exactly do you mean by that?" to a harmful question, we collected similar cases from our experiments on the Harmless dataset and obtained 110 test comparisons, where each question contains harmful or sensitive contents, one response (from the Harmless dataset) is something like "what do you mean?" and the other response (from our perturbations) adds the disrespectful phrase "the hell" into that response. We discovered that for 30\% and 15\% of the time (respectively for the two models), the two 8B RMs prefer the disrespectful response. Therefore, through our method, we found that such undesirable behaviours not only exist in the smaller RMs but also exist in the state-of-the-art ones. Details of this new analysis are written in Appendix G.

This new experiment is a good example of how our proposed explanation method enables fine-grained explorations of RM behaviour, that complement those originally included in the paper. It illustrates many potential analyses that could be performed after generating contrastive explanations, thus highlighting the potential impact and flexibility of our method.

### **2. About the list of attributes in our method.**

We kept our formalisations of contrastive explanations agnostic to the list of attributes specified. In the paper, we instantiated our list of attributes suitable for the datasets we used, and we adopted the same list for all experiments because we would like to perform global analysis along the attributes. Through our method, we could interpret and calibrate RM behaviours against the list of attributes we’re interested in, which existing methods couldn’t offer. Alternatively, the list of attributes could be easily customised, e.g., asking an LLM to identify possible relevant attributes for each comparison, thus using a different list of attributes for every comparison to explain.

We believe our work complements the existing literature on identifying various types of biases in RMs, e.g. [Zeng et al., 2024], [Park et al., 2024], [Wang et al., 2024a], allowing us to interpret RM preferences along any attributes.

---

### Meta-Review · Area_Chair_QiDp · 2024-12-28

**Metareview:**

The paper proposes a 2-step method to generating diverse completion perturbations to improve the diversity of contrastive explanations of reward model outputs. The reviewers noted that the paper is well-written and the evaluations is generally high quality.

**Additional Comments On Reviewer Discussion:**

NA

---

### Decision · Program_Chairs · 2025-01-22

Accept (Poster)